# Antimicrobial Resistance and Phylogenetic Relatedness of *Salmonella* Serovars in Indigenous Poultry and Their Drinking Water Sources in North Central Nigeria

**DOI:** 10.3390/microorganisms12081529

**Published:** 2024-07-26

**Authors:** Nancy M. Sati, Roderick M. Card, Lisa Barco, Maryam Muhammad, Pam D. Luka, Thomas Chisnall, Idowu O. Fagbamila, Giulia Cento, Nnaemeka E. Nnadi, Clovice Kankya, Innocent B. Rwego, Kokas Ikwap, Lawrence Mugisha, Joseph Erume, Frank N. Mwiine

**Affiliations:** 1College of Veterinary Medicine, Animal Resources and Biosecurity, Makerere University, Kampala P.O. Box 7072, Uganda; clovice.kankya@mak.ac.ug (C.K.); innocent.rwego@mak.ac.ug (I.B.R.); kokas.ikwap@mak.ac.ug (K.I.); lawrence.mugisha@mak.ac.ug (L.M.); joseph.erume@mak.ac.ug (J.E.); frank.mwiine@mak.ac.ug (F.N.M.); 2National Veterinary Research Institute, Vom 930103, Nigeria; maryam.muhammad@nvri.gov.ng (M.M.); idowu.fagbamila@nvri.gov.ng (I.O.F.); 3Animal and Plant Health Agency, Weybridge KT15 3NB, UK; Roderick.Card@apha.gov.uk (R.M.C.); thomas.chisnall@apha.gov.uk (T.C.); 4National and WOAH Reference Laboratory for Salmonella, Istituto Zooprofilattico Sperimentale delle Venezie (IZSVe), 10, 35020 Padova, Italy; lbarco@izsvenezie.it (L.B.); gcento@izsvenezie.it (G.C.); 5Department of Microbiology, Plateau State University, Bokkos 932111, Nigeria; eennadi@plasu.edu.ng

**Keywords:** indigenous poultry, multidrug resistant, North Central Nigeria, *Salmonella* serovars, serotyping, susceptibility testing, whole-genome sequencing

## Abstract

There is scant information on *Salmonella* in indigenous poultry in Nigeria. We investigated the occurrence and characterized *Salmonella* serovars in indigenous poultry and their drinking water sources to enhance the monitoring of the infection in poultry and to promote public health. We collected 1208 samples, poultry droppings (n = 1108), and water (n = 100) across 15 markets in North Central Nigeria. *Salmonella* spp. were isolated following World Organisation for Animal Health guidelines. *Salmonella* spp., confirmed through *inv*A gene detection by a polymerase chain reaction assay, were 6.8% (75/1108) droppings and 3% (3/100) water. Susceptibility testing against 13 antimicrobials showed 60.3% (47/78) susceptibility to all the antimicrobials tested while 14.1% (11/78) were multidrug resistant. Serotyping and whole-genome sequencing were carried out on 44 of the isolates, and 23 different serovars were identified. Genomes of serovars Luedinghausen, Laredo, Widemarsh, and Lansing are being documented in Africa for the first time. Twenty (20) antimicrobial resistance (AMR) gene markers encoding for resistance to aminoglycosides, tetracyclines, sulphonamides, quinolones, trimethoprim, penicillins and phenicols were found. Phylogenetic cluster analysis showed close relatedness among isolates from different sources. This study shows both low *Salmonella* prevalence and AMR, but since uncommon serovars are circulating, continuous monitoring is recommended so as to ensure food safety and poultry health.

## 1. Introduction

Poultry is known to harbour *Salmonella* spp., an important human foodborne pathogen, irrespective of their geographical location [1,2]. These infections could be due to host-adapted serovars of *Salmonella* spp., causing morbidity, mortality, and significant economic losses to farmers [3,4]. They may also be, as is most common, due to non-host specific zoonotic serovars which could be shed by apparently healthy poultry [5]. The shedding of *Salmonella* spp. from infected birds can cause contamination of the environment, feed, and water because it is a highly resistant microorganism that can persist for a long period after contamination [6]. 

Indigenous poultry, also known as scavenging, village, local, or backyard poultry [7], are commonly found in many African nations [8,9]. Based on the Food and Agriculture Organisation (FAO)’s farm structure categorisation, indigenous poultry are between 1 and 199 birds while commercial farms are typically 1000 or more birds [10]. Ajayi et al. [11] reported that in Nigeria, about 90% of the figure derived from indigenous poultry stock comprises chickens (91%), guinea fowls (4%), ducks (3%), and turkeys and others (2%). Indigenous poultry production provides a source of income and employment for households as well as food security and poverty alleviation for rural dwellers [11,12,13].

However, indigenous poultry production is faced with constraints, such as poor or inadequate housing and lack of disease management and control programmes [14]. The farmers/owners of these indigenous poultry have close interaction with their birds and may share common housing with the birds or provide a separate room for them [15]. This cohabitation provides an avenue for the transmission of zoonotic infections like salmonellosis and may enhance the risk of dissemination of antimicrobial resistance from such infections when they occur.

Antimicrobial resistance (AMR) represents a significant threat to global food security and human health. The burden of AMR is rising in low-, medium-, and high-income countries. These resistant phenotypes emerge due to selective pressure from prolonged and indiscriminate use of antimicrobials as feed additives and for prophylaxis. Furthermore, the rise in the global intensification of the poultry industry increases the use of antibiotics to enhance production to meet the rising demands for poultry meat and eggs. These facilitate the rise in multidrug-resistant (MDR) food-borne pathogens [16,17,18]. While there are several reports of *Salmonella* spp. infection in commercial flocks in Nigeria [19,20,21,22,23,24,25], the information about its occurrence in indigenous poultry is limited [26]. In the same vein, the frequency of isolation of *Salmonella* spp. strains resistant to antimicrobial agents continues to increase in Nigeria [1,21,27]. 

There are about 80 million indigenous poultry in Nigeria [28], and the detection of both host-adapted and non-typhoidal *Salmonella* serovars in this category of poultry needs to be better documented and not neglected. Surveillance is crucial in tracing outbreaks as well as for disease control. This study was designed to investigate the occurrence of, as well as characterize *Salmonella* spp., from indigenous poultry in North Central Nigeria to enhance the monitoring of the infection in poultry and, ultimately, to protect public health. 

## 2. Materials and Methods

### 2.1. Study Area and Design

The study was conducted in three out of the six North Central States (Benue, Kwara, and Plateau) of Nigeria. Fifteen markets from the three North Central States of Nigeria were randomly selected by balloting without replacement as sampling sites for this study. The locations of these markets are shown in Figure 1.

### 2.2. Sampling Size

Since the prevalence rate for *Salmonella* spp. infections in indigenous poultry was not known, an assumed prevalence rate of 50% was taken to calculate the sample size using the formula by Thrusfield [29] with a margin of error of 5% and a confidence level of 95%. A total of 385 samples were required per State to give a sample size of 1155 (n = 385 × 3).

### 2.3. Ethical Approval 

This study was approved by the Animal Use and Care Committee (AUCC) of the National Veterinary Research Institute Vom, Nigeria, with Reference number AEC/02/70/19.

### 2.4. Sampling Locations and Sample Collection 

From each selected State, five markets were selected as sampling locations for this study: Yandev, Gboko, Wurukum, Otukpo, and Modern markets in Benue State; Sango, Oja-Oba, Share, Ipata, and Oke-Oyi markets in Kwara State; Bokkos, Mangu, Shendam, Yankaji, and Kugiya markets in Plateau State (Figure 1). Sampling was carried out in each selected market over two visits at least three months apart. Some of these markets are conducted daily, weekly, or on alternate days. Markets were used as sampling sites because they serve as congregating points where farmers from nearby villages bring their products to a central place where people can come and buy such products. Additionally, farmers are also likely to bring healthy live birds for sale to the markets, and such birds were the target group for the sampling in this study. In each of the markets visited, freshly voided poultry droppings (5 g each) were collected from individual birds. Water samples (5 mL each) were also collected. Poultry droppings samples collected were considered representative of the *Salmonella* status of the birds where they were brought from. Water samples were collected from the source of supply in the markets, which served both humans and animals within each market sampled. Water sources at the markets included surface water such as streams, ponds, and rivers, or ground water such as wells and bore holes. A few markets had pipe-borne water. These water sources were sampled because of the poor hygiene practices frequently observed at water sources at markets where poultry are sold. About 40 samples were collected from each market on each of the two visits, with the total market sample size ranging between 79 and 83. Samples were properly labelled and transported on ice in cold boxes to the Bacteriology Laboratory at the National Veterinary Research Institute, Vom, Nigeria. 

### 2.5. Salmonella spp. Isolation and Identification

Isolation and identification of *Salmonella* spp. isolates were carried out according to the World Organisation for Animal Health (WOAH) guidelines [30] for isolating *Salmonella* spp., using Buffered Peptone Water for pre-enrichment and Rappaport-Vassiliadis broth and Muller Kauffman tetrathionate broth enriched with novobiocin for selective enrichment. Colonies characteristic of *Salmonella* spp., grown on xylose lysine deoxycholate media, were picked, purified, and then biochemically identified using citrate, indole, oxidase, urease, catalase, triple sugar iron, lactose, glucose, mannitol, dulcitol, sucrose, and raffinose tests as described by Hassanein et al. [31]. Media used for this experiment were sourced from (Oxoid, Basingstoke, Hampshire, UK). The prevalence of *Salmonella* (proportion of positive samples) and the 95% confidence interval of the proportions were calculated with GraphPad using the modified Wald method for confidence intervals (https://www.graphpad.com/quickcalcs/confInterval1/ (accessed on 12 June 2024).

### 2.6. Polymerase Chain Reaction confirmation of Salmonella spp.

Phenotypically identified *Salmonella* spp. isolates were confirmed by conventional polymerase chain reaction (PCR) assay. Genomic DNA extraction was carried out using DNeasy Blood and Tissue kit (Qiagen, Hilden, Germany) according to the manufacturer’s instructions. The assay utilized appropriate primers (Forward: 5′-GTG TTA TCG CCA CGT TCG GGC AA-3′ and Reverse: 5′-TCA TCG CAC CGT CAA AGG AAC C 3′) that targeted *inv*A gene fragment and PCR cycling conditions as described by Rahn et al. [32]. The *inv*A gene is targeted because it is found in all known serovars of *Salmonella* spp. [33,34]. 

### 2.7. Antimicrobial Susceptibility Testing

The susceptibility of *Salmonella* spp. isolates against a panel of 13 antimicrobials that are commonly used in treating salmonellosis in animals and humans was tested using the disk diffusion method as described by Bauer et al. [35]. The antimicrobials used were azithromycin (AZM) 15 µg, levofloxacin (LEV) 5 µg, ceftazidime (CAZ) 30 µg, cefotaxime (CTX) 30 µg, gentamicin (CN) 10 µg, amikacin (AK) 30 µg, meropenem (MEM) 10 µg, nitrofurantoin (F) 300 µg, sulphamethoxazole-trimethoprim (SXT) 23.75 µg + 1.25 µg, tetracycline (TE) 30 µg, tigecycline (TG) 15 µg, ampicillin (AMP) 10 µg, and chloramphenicol (C) 30 µg. The Clinical and Laboratory Standards Institute (CLSI) and European Committee on Antimicrobial Testing (EUCAST) guidelines were followed, and the clinical breakpoints were employed as interpretive criteria for 11 and 2 antimicrobial agents, respectively, to classify bacterial isolates as Susceptible, Intermediate, or Resistant [36,37]. Multidrug-resistant *Salmonella* spp. isolates were categorised as resistant to at least one antimicrobial agent in three or more classes of antimicrobials [38].

### 2.8. Multiple Antimicrobial Resistance Indexing

The multiple antimicrobial resistance (MAR) index, as described by Krumperman et al. [39], was also determined by calculating the ratio of antimicrobials an isolate is resistant to and the total number of antimicrobials the isolate was exposed to. A value greater than 0.2 is indicative of a high-risk source of contamination where antimicrobials are often used [40].

### 2.9. Serotyping of Isolates and Whole-Genome Sequencing

Isolates that were confirmed as *Salmonella* spp. via *inv*A gene detection were sent to Istituto Zooprofilattico Sperimentale delle Venezie (IZSVe), Padova, Italy, and to Animal and Plant Health Agency (APHA), Weybridge, UK, for serotyping and sequencing. Serotyping was carried out by identification of surface O antigens and flagella H antigens. 

For WGS, briefly, pure cultures of *Salmonella* spp. isolates were grown overnight on Tryptose agar at 37 °C. The extraction of genomic DNA (gDNA) was performed using a commercial kit (QIAamp DNA Mini QIAGEN, (Milan, Italy)), and pure gDNA was quantified with a Qubit 3.0 Fluorometer (Life Technologies, Waltham, MA, USA). Libraries were prepared using the Nextera XT DNA sample preparation kit (Illumina, San Diego, CA, USA). Miseq Reagent kitv3 was used to perform high-throughput sequencing, resulting in 251 bp long paired-end reads. FastQC v0.11.2 [41] was used to assess the sample quality, while Trimmomatic 0.32 [42] was used to trim the quality and length. Nextera adapter sequences were removed by cutting bases off the reads. Reads were subsequently assembled de novo using Spades 3.10.1 [43] with default parameters for Illumina reads, and the quality of assembly was assessed using QUAST 3.1 [44]. The presence of genes and point mutations conferring AMR, heavy metal stress genes, and virulence genes were assessed using AMRFinderPlus v3.12.8 [45]. The Sequence Type (ST) was determined with MLST [version 2.19.0] [46]; using the pubMLST database (https://pubmlst.org) accessed on 5 June 2024 [47].

### 2.10. Multilocus Sequence Typing and Cluster Analysis

Core genome multilocus sequence typing (cgMLST) and cluster analysis were carried out to reveal genetic relationships within each serovar across the different sources and markets sampled. These were performed using the gene-by-gene approach, whereby MLST analysis was upscaled to include thousands of genes or parts of genes [48]. For *Salmonella enterica*, a total of 3255 loci, using the INNUENDO cgMLST scheme, were investigated https://doi.org/10.5281/zenodo.1323684 accessed on 28 May 2024 [49].

### 2.11. Data Availability

The genomes of the 44 isolates have been deposited to NCBI under the bioproject PRJNA941491 with accession numbers (JBEQHX000000000-JBEQJK000000000, JARGCR000000000, JARKIQ000000000, JBEVCX000000000 and JBEVCY000000000). This is attached as a Appendix A (Appendix A).

## 3. Results

### 3.1. Prevalence of Salmonella spp. in Indigenous Poultry and Poultry Drinking Water Sources in the Markets 

The overall prevalence of *Salmonella* in indigenous poultry was 6.8% (confidence interval 5.4 to 8.4%), comprising 75 positives from 1108 poultry droppings (Appendix A). *Salmonella*-positive poultry were detected at all 15 markets assessed in this study, and prevalence ranged from 3.9% at Bokkos market in Plateau State to 10.5% at Shendam market in Plateau State (Appendix A). *Salmonella* was detected in the water sources at three of the 15 markets (Otukpo and Modern markets in Benue State; Oke-Oyi market in Kwara State), with one positive sample at each market from a total of 100 water samples tested, giving an overall prevalence of 3.0% (confidence interval 0.65 to 8.8%) (Appendix A).

### 3.2. Antimicrobial Susceptibility Profiles of Isolates from the Sampled Markets 

All the 78 isolates were susceptible to meropenem and tigecycline (Figure 2). The isolates exhibited no resistance to amikacin, except one isolate that displayed an intermediate resistance phenotype. A low occurrence of resistance was noted for ceftazidime (2/78), nitrofurantoin (2/78), levofloxacin (3/78), cefotaxime (3/78), and gentamicin (4/78). On the other hand, antimicrobials with a higher occurrence of resistance were chloramphenicol (6/78), azithromycin (6/78), sulphamethoxazole-trimethoprim (9/78), tetracycline (13/78), and ampicillin (16/78). 

### 3.3. Spectrum of Antimicrobial Resistance of the Salmonella spp. Isolates

Table 1 shows the Multiple Antimicrobial Resistance (MAR) index which ranged from 0.08 to 0.46. Among these, 14.1% (11/78) had an MAR of >0.2. Antimicrobial resistance was seen in 39.7% (31/78) of the isolates while 60.3% (47/78) were fully susceptible to all the antimicrobials tested. Single resistance against one particular antimicrobial was noted in 17.9% (14/78) of the isolates. Double resistance (resistance to two antimicrobials) was detected in 7.7% (6/78) of the isolates. The MDR *Salmonella* spp. isolates were 14.1% (11/78) and had different patterns of antimicrobial resistance with resistance to Chloramphenicol, Sulphamethoxazole-trimethoprim, and tetracycline being the most dominant occurring in 27.3% (3/11) of the isolates. The MDR isolates were resistant to between three and six different antimicrobials.

### 3.4. Distribution Patterns of Antimicrobial Resistance from Different Markets against the 13 Antimicrobials

Different types of antimicrobial resistance distribution patterns were seen across the different isolates at the markets. Table 2 summarizes the different patterns of AMR seen in each of the markets sampled. The antimicrobial resistance of the isolates from the five markets in Benue State against the 13 antimicrobials ranged from 0% to 53.8%. Isolates from Otukpo market in Benue State exhibited resistance to 53.8% (7/13) of the antimicrobials tested, while there was none recorded in Wurukum market. In Kwara State, the highest resistance was in Oke-Oyi market, with resistance occurring against 38.5% (5/13) antimicrobials and Share market having none. Similarly, Plateau State had the highest resistance of 38.5% (5/13) at Yankaji market and the least at Mangu and Shendam markets, with 7.7% (1/13) each.

### 3.5. Salmonella spp. Serovars Detected from Indigenous Poultry and Poultry Drinking Water Sources from Markets 

Out of the 78 isolates from this study, 44 isolates were selected from the 15 markets for serotyping and whole-genome sequencing. This included all isolates from water and other poultry species outside chicken (water 3, duck 3, and turkey 1), in addition to 37 from chicken sources. This was to ensure representation of all samples collected. Twenty-three different serovars were identified (Appendix A). *Salmonella* Chester had the highest number of isolates (n = 6), followed by serovars Isangi and Agama (n = 4 each), and serovars Offa and Derby (n = 3) isolates each. Six serovars had two isolates each: serovars Saintpaul, Laredo, Give, Orion, Monophasic Variant of *Salmonella* Typhimurium, and Widemarsh. The remaining 12 serovars each had one single isolate: serovars Typhimurium, Telelkebir, Durham, Larochelle, Kingston, Vom, Linguere, Bareilly, Lansing, Luedinghausen, *Salmonella enterica* subspp *enterica* 6,7:c−, and *Salmonella enterica* subspp. *enterica* 6,7:a−. The three isolates from water samples comprised two *Salmonella* Chester and one *Salmonella* Derby. Figure 3 shows the distribution of the serotyped strains (44) isolated from the 15 markets across the three States sampled.

### 3.6. Detection of Novel Sequence Types, Antimicrobial Resistance Gene Markers and Their Distribution in Salmonella serovars

The 44 strains of *Salmonella* spp. sequenced were screened for the presence of antimicrobial gene markers. We detected floR, *tet*A and *bla*TEM-215 genes in some of the serovars in this study that confer resistance against phenicol, tetracyclines, and penicillins, respectively. Resistance against quinolones was established by the detection of *qnr*B1, *qnr*B5, *qnr*B13, and *qnr*B19 genes. Similarly, resistance to sulphonamides/trimethoprim was encoded by 6 gene markers: *sul1*, *sul*2, *sul*3, *dfr*A1, *dfr*A15, and *dfr*A17. Resistance against aminoglycosides was the highest being encoded by eight different gene markers: *aac*(3)-Via, *aph*(3)-Ia, *aph*(3)-Ib, *aph*(6)-Id, *aac*(3)-Ile, *aad*A1, *aad*A5, f*os*A7. 

Table 3 shows the phenotypic antimicrobial resistance as well as the distribution of antimicrobial gene markers in the 44 serovars. Phenotypically and genotypically, *S.* Isangi strains showed similar results for chloramphenicol, sulphamethoxazole-trimethoprim, and tetracycline. However, there was no phenotypic resistance for quinolones, but the gene was detected. Additionally, there was no resistance to aminoglycosides in the disc diffusion results, yet AMR gene markers for this class of antimicrobials were detected. Quinolone resistance genes were expressed in some of the *S.* Agama strains, but the antimicrobial susceptibility testing showed no resistance.

Some serovars including Chester, Agama, Offa, Widemarsh, Orion, Saintpaul, Laredo, and MVST had similar antimicrobial gene markers, irrespective of market of origin. Two of the Derby strains from the same market bore seven identical antimicrobial gene markers suggesting the strains are closely related in terms of antimicrobial gene markers. Four *Salmonella* Isangi strains were identified, two of which originated from Gboko market while the third and fourth were from Yandev and Bokkos markets, respectively. One Isangi serovar from Yandev market had two antimicrobial gene markers, while the two strains from Gboko market had seven and ten gene markers respectively. The Isangi serovar from Bokkos also had 10 antimicrobial gene markers, all of which were identical to the one from Gboko, suggesting these strains to be closely linked. Some of the serovars bore one antimicrobial gene marker while others had none. 

From this study, we identified 23 different *Salmonella* sequence types, out of which 12 are novel and have been uploaded on EnteroBase (https://enterobase.warwick.ac.uk/, accessed on 12 July 2024). 

### 3.7. Detection of Virulence Genes

The strains were run through the NCBI AMRFinder tool, and the cytolethal distending toxin B gene *cdt*B was present in the *S.* Chester, *S.* Durham, *S.* Give, *S.* Laredo, *S.* Telelkebir, and *S.* Vom isolates. One monophasic *S.* Typhimurium isolate (23-43791_S8) harboured the superoxide dismutase *SodCI*.

### 3.8. Detection of Rare Salmonella serovars and Their EnteroBase Statuses

Some uncommon serovars of *Salmonella* were detected in this study as shown in Table 4. 

### 3.9. Phylogenetic Relatedness of the Salmonella Isolates

The cgMLST cluster analysis of isolates of *Salmonella* serovars Widemarsh, Orion, Offa, Laredo, Isangi, Derby, and Agama obtained from different markets and sources in the three states showed a close relationship among them. In contrast, cgMLST cluster analysis between the two MVST strains isolated from poultry droppings of chickens in Mangu and Modern markets in Plateau and Benue States, respectively, revealed the lack of genetic correlation between them. Finally, in relation to *S.* Chester isolates, five out of the six from three different markets and sources across two States were closely related, whereas another single isolate showed lack of genetic correlation to the others. These relationships are shown in attached supplementary figures (Appendix A).

Figure 4 describes the interesting distribution of the strains based on the markets sampled with the Share and Modern markets having the highest number of strains, although some belong to the same serovar but may be from different sources and species. The distribution of sequences shows that they are influenced by the serovar, and isolates belonging to the same serovar, obtained from different markets, showed closely related sequences (e.g., *S.* Agama from three markets in Plateau State; *S.* Chester from three markets, one in Benue and two in Kwara States).

## 4. Discussion

To our knowledge, this is the first cross-sectional market-based study to investigate the occurrence of *Salmonella* and its antimicrobial resistance in indigenous poultry and poultry drinking water sources in North Central Nigeria. Our findings show the presence of *Salmonella* spp. in the samples across the 15 markets tested. There was a prevalence of 6.8% (75/1108) of *Salmonella* spp. in poultry faeces and 3.0% (3/100) from the water sampled. Specific market prevalence ranging between 3.9 and 10.5% bears great significance and highlights salmonellosis to be endemic in indigenous poultry in this region. These findings are reinforced by the detection of 23 different *Salmonella* spp. serovars, implying there is a potential risk of exposure of the public to *Salmonella* serovars from indigenous poultry in this region of Nigeria or vice versa, and the circulating strains are very diverse and heterogeneous. Additionally, some of the serovars such as *S.* Lansing*, S.* Laredo, and *S.* Widemarsh are not commonly isolated from poultry sources. Indigenous poultry are by nature scavengers and may probably pick up *Salmonella* serovars from the environment, which are not the typical ones isolated from poultry reared using more intensive farming practices. The need to ensure that only wholesome products are produced along the food value chain must be prioritized as consumer preference for indigenous poultry and their products in Nigeria continues to increase, which may lead to increase in cases of nontyphoidal infections.

Our findings are consistent with those of others outside Nigeria [50,51,52]: the prevalence of *Salmonella* infections in free-range or indigenous poultry is lower than in commercial or intensively raised poultry. Similarly, within Nigeria, our study agrees with the report of Salihu et al. [53] in Nasarawa state, who reported 8.5% seroprevalence and a 2.5% prevalence [54] of *Salmonella* Gallinarum in free-range chickens. A *Salmonella* spp. prevalence rate of 10% in local chickens retailed along the roadside in Zaria was reported by Ejeh et al. [55]. The low prevalence rate of *Salmonella* infections obtained in this research may be due to the hardiness of indigenous poultry and may also be because apparently healthy birds were sampled. *Salmonella* prevalence rates of 11% and 27% have been reported in commercial poultry in Ibadan and Maiduguri, Nigeria, respectively [56,57]. Fagbamila et al. [24] reported 43.6% in commercial poultry farms in Nigeria with state prevalence ranging from 11.1 to 65.4%, and more recently, in North-Western Nigeria, by Jibril et al. [25] who reported a prevalence of 47.9% in commercial poultry. The higher prevalence in commercial poultry may be due to the bigger flock sizes, leading to a higher probability of occurrence, persistence, and spread of *Salmonella* infections [58]. 

Majority of the isolates (60.3%) were fully susceptible to all the antimicrobials tested, which concurs with the findings of Alhaji et al. [59], who reported that 92.1% of farmers of free-range poultry in North Central Nigeria rarely use antimicrobials. Since indigenous poultry are known to be kept in small flock sizes [26], this does not encourage the farmers to invest in treatment by incurring additional costs on drugs needed to treat infections when they arise, as farmers would generally opt for slaughter. In addition, the near absence of veterinary or extension services for diagnosis and treatment also does not favour the use of antimicrobials to treat indigenous poultry. These might account for the low occurrence of antimicrobial resistance detected in this study. On the other hand, in Nigeria, commercial poultry farmers regularly use antimicrobials for growth enhancement, prophylaxis, or therapy [60,61], which is associated with higher antimicrobial resistance [62]. 

The presence of AMR observed in some isolates obtained in this study, poses potentially dangerous health and economic implications for poultry and human health. High resistance to tetracycline in commercial poultry has been reported by Agbaje et al. [63], and a survey conducted by Fasina et al. [64] ranked tetracycline-sulphonamide resistance as one of the most frequently occurring. Although there are hardly any data on antimicrobial resistance in indigenous poultry in Nigeria, this study has shown that indigenous poultry have lower antimicrobial resistance compared to commercial poultry. Interestingly, some of the isolates that did not exhibit antimicrobial resistance phenotypically had AMR genes after genotypic testing was performed. However, the genotypic detection of AMR genes does not always mean the genes detected will fall within the intermediate or resistant interpretive category [65]. Similarly, our study is in agreement with that of Deekshit and Srikumar [66] who reported that some genes maybe “silent” and not necessarily be associated with any corresponding phenotypic expression. 

From the 15 markets sampled, 23 different serovars of *Salmonella* were isolated. All the 23 serovar types were found in poultry droppings while two types (serovars Derby and Chester) were found in water. Chester was the most prevalent serovar (6/44) followed by Agama and Isangi (4/44). Serovars identified in this study such as Chester, Isangi, Give, Larochelle, Agama, Durham, and Telelkebir have also been isolated from commercial poultry in different parts of Nigeria [25,63]. Similarly, Ikhimiukor et al. [27] isolated Chester, Durham, Dublin, Telelkebir, and Typhimurium from humans in southwestern Nigeria. The detection of some virulence genes (*cdtB* gene and *SodCI*) in some serovars indicates that these serovars are potential threats to the health of the public. Our findings suggest that these serovars are also circulating widely in Nigeria’s indigenous poultry. Thus, deliberate efforts must be made to ensure effective food hygiene measures are enforced to reduce risk to consumers. Moreover, this study showed that in indigenous poultry in Nigeria, some rare serovars, which have been rarely described before, are circulating, so this calls for regular monitoring of indigenous poultry so that these pathogens are kept under check to avoid salmonellosis. The study has also enriched our gene database and given us more insights into the genomic diversity that exists in *Salmonella* serovars. To the best of our knowledge, going through EnteroBase [67], this is the first report of WGS of some serovars from Africa. *Salmonella* Widemarsh has 32 submissions but none from Africa. *Salmonella* Lansing has 12 submissions, and none is from Africa. Additionally, *Salmonella* Luedinghausen and *Salmonella* Laredo both had only one WGS in EnteroBase prior to this study.

The cgMLST showed the span of the relatedness and diversity of the isolates in this study. Isolates belonging to different serovars were investigated based on source, location, and species. Serovars Widemarsh, Orion, and Laredo isolated from poultry droppings from two markets in the same State were closely related. Close antigenic relationship among isolates from different markets and states suggests that the isolates share a common ancestry and shows how these pathogens are transmitted across species and the environment. The same applies to *Salmonella* Offa isolated from chicken and duck droppings in two markets having a close relationship, meaning the isolates circulating around that area have an establishment of ecological niches resulting in geographical clustering of serovars [68]. The same situation was also found for *Salmonella* Agama and *Salmonella* Isangi isolated from different sources from three markets in two and one states, respectively. Serovar MVST isolated from two markets in different States reveal no close relationship as it was for *Salmonella* Give isolated from poultry faeces in two markets from two different states. *Salmonella* Derby is obviously circulating in Modern Market as it was found in water and faeces, and the isolates show a close relationship. Interestingly, *Salmonella* Chester isolates exhibited a clustering of five (from water and faeces across different markets in different States) out of the six having common and close ancestors and the last isolate being distinctly distant. These findings show epidemiological dimensions that need to be explored further.

### Limitations

Seventy-eight *Salmonella* isolates were obtained from this study. However, due to logistic reasons, 44 where serotyped and sequenced. The *Salmonella* reference laboratory, IZSVe, Padova, Italy, worked on 42 isolates, while out of the 6 sent to APHA, Weybridge, UK, only 2 isolates were viable. 

## 5. Conclusions

This study has provided baseline data on *Salmonella* prevalence in indigenous poultry and their drinking water sources in North Central Nigeria, suggesting that indigenous poultry in Nigeria have lower *Salmonella* prevalence as well as less AMR compared to commercial poultry. However, these poultry do harbour a wide diversity of potentially zoonotic *Salmonella* strains that are not seen in other food sources, and as they are widely consumed, it will be important to continue to monitor this, especially as some of the serovars have rarely been described before, indicating that the investigation of their phylogeny and epidemiology deserves further effort. To be able to improve poultry productivity and ensure food safety, which will promote the health of the public, diseases such as salmonelloses have to be studied, understood, and controlled at the animal, environmental, and human interphase levels.

## Figures and Tables

**Figure 1 microorganisms-12-01529-f001:**
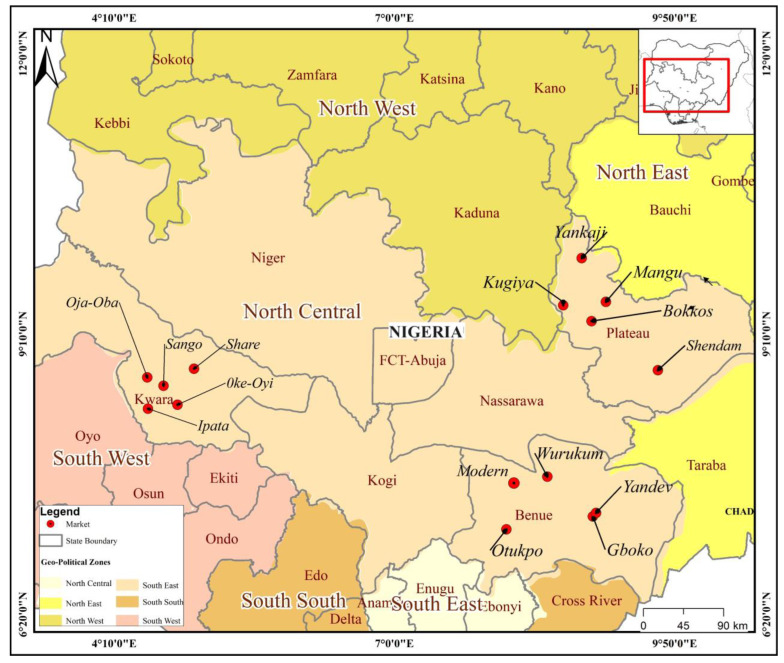
Map showing the 15 markets in North Central Nigeria where indigenous poultry faeces and water samples were collected.

**Figure 2 microorganisms-12-01529-f002:**
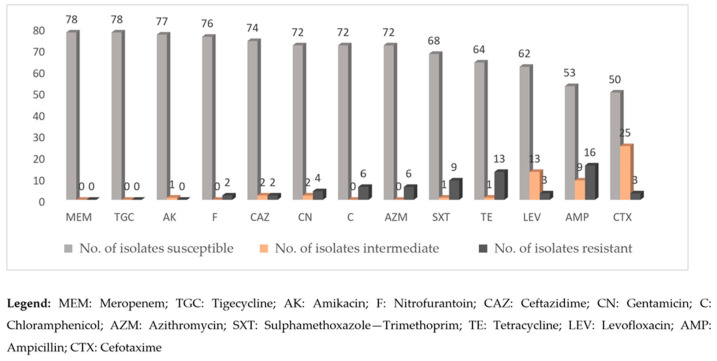
Antimicrobial susceptibility patterns of *Salmonella* spp. strains (78) isolated from indigenous poultry and poultry drinking water sources in 15 markets in North Central Nigeria.

**Figure 3 microorganisms-12-01529-f003:**
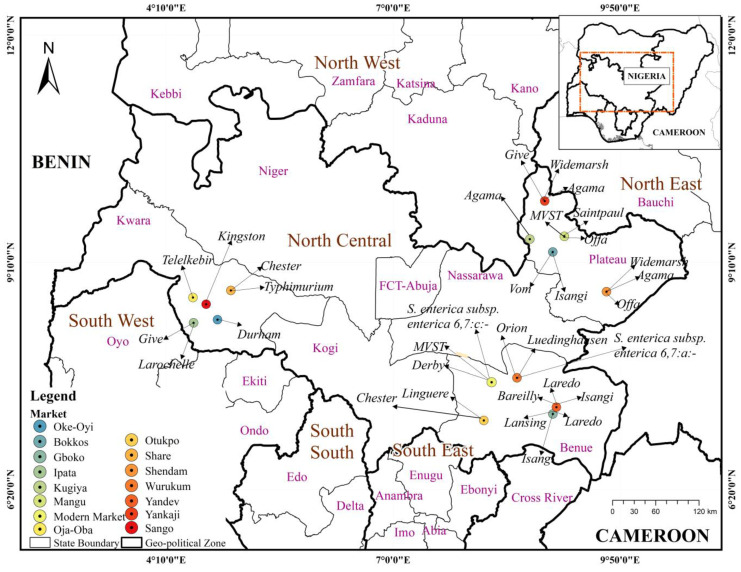
Map showing the *Salmonella* serovars found in the 15 markets sampled in North Central Nigeria.

**Figure 4 microorganisms-12-01529-f004:**
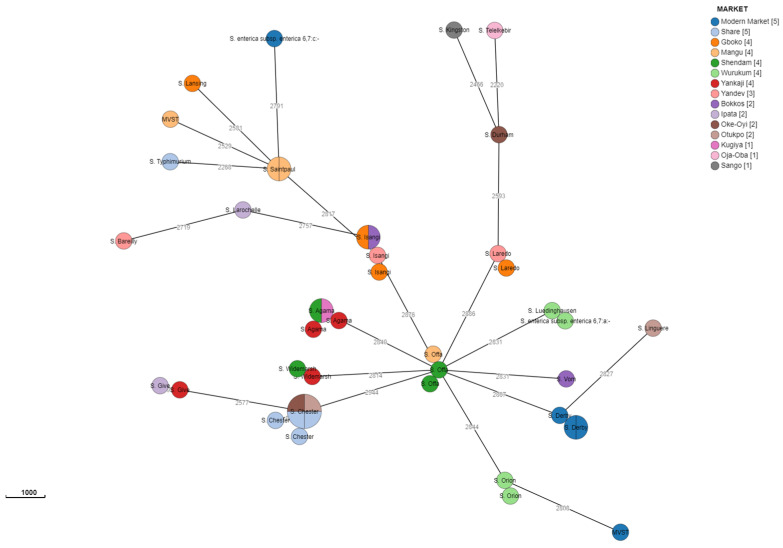
Cluster analysis for the 44 strains based on the markets sampled.

**Table 1 microorganisms-12-01529-t001:** Antimicrobial resistance spectrum and Multiple Antimicrobial Resistance (MAR) Index of *Salmonella* spp. isolates (Isolates susceptible to all the 13 antimicrobials tested were defined as fully susceptible).

Resistance Profiles	No of Isolates	MAR Index
**Fully susceptible**		
**Total**	**47**	≤0.2
**Percentage**	**47/78 (60.2%)**	
**Single resistance**		
AZM	1	≤0.2
AMP	5	≤0.2
CTX	3	≤0.2
TE	2	≤0.2
SXT	2	≤0.2
CN	1	≤0.2
**Total**	**14**	
**Percentage**	**14/78 (17.9%)**	
**Double resistance**		
AMP, TE	1	≤0.2
AZM, AMP	2	≤0.2
C, TE	1	≤0.2
AMP, CAZ	1	≤0.2
AMP, LEV	1	
**Total**	**6**	
**Percentage**	**6/78 (7.7%)**	
**MDR**		
C, SXT, TE	3	≥0.2
CN, AMP, TE	1	≥0.2
SXT, AMP, TE	1	≥0.2
SXT, TE, CAZ	1	≥0.2
F, AZM, TE	1	≥0.2
C, SXT, AMP	1	≥0.2
CN, C, AMP, LEV	1	≥0.2
F, AZM, AMP, TE	1	≥0.2
CN, LEV, AZM, SXT, AMP, TE	1	≥0.2
**Total**	**11**	
**Percentage**	**11/78 (14.1%)**	

Legend: AMP: Ampicillin, AZM: Azithromycin, CTX: Cefotaxime, TE: Tetracycline, CN: Gentamicin SXT: Sulphamethoxazole-Trimethoprim, C: Chloramphenicol, CAZ: Ceftazidime, LEV: Levofloxacin, F: Nitrofurantoin.

**Table 2 microorganisms-12-01529-t002:** Phenotypic antimicrobial resistance in the *Salmonella* spp. from the 15 markets sampled.

State	Market	Antimicrobial Resistance Types	Number of Isolates	Antimicrobial Resistance Type No. (%)
Benue	Otukpo	CN, C, LEV, AZM, SXT, AMP, TE	6	7/13 (53.8)
	Modern Market	CTX, CN, AMP, TE	4	4/13 (30.8)
	Wurukum	None	0	0/13 (0)
	Gboko	C, SXT, TE	2	3/13 (23.1)
	Yandev	SXT	1	1/13 (7.7)
Kwara	Sango	AZM, AMP	2	2/13 (15.4)
	Oja-Oba	CTX, AMP	2	2/13 (15.4)
	Ipata	AMP	1	1/13 (7.7)
	Share	None	0	0/13 (0)
	Oke-Oyi	C, AZM, SXT, AMP, TE	4	5/13 (38.5)
Plateau	Mangu	AMP	1	1/13 (7.7)
	Bokkos	C, SXT, TE	2	3/13 (23.1)
	Shendam	TE	1	1/13 (7.7)
	Yankaji	CTX, SXT, AMP, TE, CAZ	3	5/13 (38.5)
	Kugiya	F, AZM, AMP, TE	2	4/13 (30.8)

Legend: AMP: Ampicillin, AZM: Azithromycin, CTX: Cefotaxime, TE: Tetracycline, CN: Gentamicin SXT: Sulphamethoxazole-Trimethoprim, C: Chloramphenicol, CAZ: Ceftazidime, LEV: Levofloxacin, F: Nitrofurantoin.

**Table 3 microorganisms-12-01529-t003:** Characteristics of *Salmonella* spp. isolated from indigenous poultry in markets located in Northern Nigeria: Phenotypic profiles and genotypic markers of antimicrobial resistance, serovars, and sequence types.

Market	Laboratory ID No	Serovar	Sample Source	Sequence Type	Phenotypic Antimicrobial Resistance	Antimicrobial Resistance Genes
Otukpo	23-43794_S15	*S.* Chester	Water	411	None	None
Share	23-43796_S16	*S.* Chester	Chicken	411	Azithromycin	None
Share	23-43797_S10	*S.* Chester	Chicken	411	None	None
Share	23-43799_S11	*S.* Chester	Duck	411	None	None
Share	23-43801_S12	*S.* Chester	Duck	411	None	None
Oke-Oyi	23-43802_S13	*S.* Chester	Water	411	None	None
Yandev	23-43829_S2	*S.* Isangi	Chicken	216	Sulphamethoxazole-trimethoprim	*dfr*A17, *aad*A5, *aph*(3′)-Ia, *sul*1, *qnr*B19
Gboko	23-43828_S1	*S.* Isangi	Chicken	216	Chloramphenicol Sulphamethoxazole-trimethoprim Tetracycline	*tet*(A), *floR*, *aac*(3′)-Via, *dfr*A15, *qnr*B19,*dfr*A17, *aad*A5, *aad*A1, *sul*1, *aph*(3′)-Ia
Bokkos	IFSO 23	*S.* Isangi	Chicken	216	Chloramphenicol Sulphamethoxazole-trimethoprim Tetracycline	*floR*, *tet*(A*), aac*(3)-Via, *aad*A1, *dfr*A15, *qnr*B19, *sul*1, *aph*(3′)-Ia, *aad*A5, *dfr*A17
Gboko	IFSO 25	*S.* Isangi	Chicken	216	Chloramphenicol Sulphamethoxazole-trimethoprim Tetracycline	*floR*, *tet*(A*), aac*(3′)-Via , *aad*A1, *dfr*A15 , *qnr*B19, *sul*1, *aph*(3′)-Ia,, *qnr*B19, *aad*A5, *dfr*A17
Yankaji	23-43807_S19	*S.* Agama	Chicken	11,508 *	Ampicillin Ceftazidime	*qnr*B19
Kugiya	23-43808_S20	*S.* Agama	Turkey	11,508 *	None	*qnr*B19
Shendam	23-43809_S21	*S.* Agama	Chicken	11,508 *	None	*qnr*B19
Yankaji	23-43810_S22	*S.* Agama	Chicken	11,508 *	None	*qnr*B19
Modern market	23-43822_S27	*S.* Derby	Chicken	9580	None	None
Modern market	23-43821_S26	*S.* Derby	Chicken	9580	Gentamicin Ampicillin Tetracycline	*bla*_TEM-215_, *qnr*S13, *tet*(A), *aph*(3″)-Ib, *aph* (6′)-Id, *sul*2, *aac*(3′)-Ile
Modern market	23-43819_S25	*S.* Derby	Water	9580	Gentamicin	*bla*_TEM-215_, *qnr*S13, *tet*(A), *aph* (3″)-Ib, *aph*(6′)-Id, *sul*2, *aac*(3′)-Ile
Mangu	23-43786_S13	*S.* Offa	Chicken	11,457 *	None	None
Shendam	23-43788_S14	*S.* Offa	Chicken	11,457 *	None	None
Shendam	23-43790_S7	*S.* Offa	Duck	11,457 *	None	None
Yankaji	23-43804_S17	*S.* Widemarsh	Chicken	11,452 *	Cefotaxime	None
Shendam	23-43805_S18	*S.* Widemarsh	Chicken	11,452 *	None	None
Yankaji	23-43780_S11	*S.* Give	Chicken	516	Sulphamethoxazole-trimethoprim Tetracycline Ceftazidime	*dfr*A1, *tet*(A), *sul*3, *qnr*S1, *aad*A1
Ipata	23-43783_S12	*S.* Give	Chicken	516	None	None
Wurukum	23-437825_S28	*S.* Orion	Chicken	11,455 *	None	None
Wurukum	23-43827_S29	*S.* Orion	Chicken	11,455 *	None	None
Mangu	23-43813_S23	*S.* Saintpaul	Chicken	11,456 *	None	*fos*A7
Mangu	23-43815_S24	*S.* Saintpaul	Chicken	11,456 *	None	*fos*A7
Gboko	23-43833_S3	*S.* Laredo	Chicken	11,458 *	None	None
Yandev	23-43834_S4	*S.* Laredo	Chicken	11,458 *	None	None
Mangu	23-43791_S8	MVST	Chicken	19	None	None
Modern Market	23-43793_S9	MVST	Chicken	11,459 *	Ampicillin Tetracycline	*fos*A7
Wurukum	23-43836_S5	*S.* Luedinghausen	Chicken	11,453 *	None	None
Sango	23-43848_S10	*S.* Kingston	Chicken	3670	None	None
Bokkos	23-43856_S4	*S.* Vom	Chicken	11,451 *	None	None
Ipata	23-43853_S2	*S.* Larochelle	Chicken	22	None	None
Oja-Oba	23-43851_S1	*S.* Telelkebir	Chicken	3326	None	None
Oke-Oyi	23-43854_S3	*S.* Durham	Chicken	2010	Azithromycin	None
Yandev	23-43844_S8	*S.* Bareilly	Chicken	11,450 *	None	None
Share	23-43847_S9	*S.* Typhimurium	Chicken	513	None	*fos*A7
Otukpo	23-43839_S6	*S.* Linguere	Chicken	11,454 *	Tetracycline	None
Gboko	23-43842_S7	*S.* Lansing	Chicken	8706	None	*fos*A7
Modern Market	23-43857_S5	*S.* 6,7:c−	Chicken	11,513 *	None	None
Wurukum	23-43858_S6	*S.*6,7:a−	Chicken	11,453 *	None	None

MVST: Monophasic Variant of *Salmonella enterica* subsp *enterica* Typhimurium. *S.* 6,7:c−: *Salmonella* enterica subsp *enterica* 6,7:c−. *S.* 6,7: a−: *Salmonella enterica* subsp *enterica* 6,7:a−. * New sequence types identified in this study.

**Table 4 microorganisms-12-01529-t004:** Some uncommon *Salmonella* serovars from this study with WGS data in EnteroBase (as at 25 May 2024).

Serovar	WGS Status
*S.* Luedinghausen	First in Africa, second ever
*S.* Laredo	First in Africa, second ever
*S.* Widemarsh	First in Africa
*S.* Lansing	First in Africa
*S.* Linguere	Second in Africa, third ever

## Data Availability

The original data presented in the study are openly available in GenBank Genome-BioProject at www.ncbi.nlm.nih.gov/bioproject/?term=PRJNA941491.

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
