# Peer review of "Antimicrobial Resistance and Phylogenetic Relatedness of Salmonella Serovars in Indigenous Poultry and Their Drinking Water Sources in North Central Nigeria"

_microorganisms, 2024, doi:10.3390/microorganisms12081529_

Round 1

Reviewer 1 Report

Comments and Suggestions for Authors

The study by Sati and colleagues reports: i) The prevalence of Salmonella in the feces of indigenous poultry and their drinking water sources; ii) The antimicrobial susceptibility profile of all Salmonella isolates; iii) The serovars of most isolates. In addition, whole genome sequencing of the majority of isolates (n=44) was performed and analyzed regarding the sequence types (MLST), and genetic markers of antimicrobial resistance and virulence.

Overall the study is technically sound, and the conclusions are supported by the results presented. However, some questions need to be clarified:

-In lines 86 and 87, the authors describe that the markets included in the analysis “were randomly selected by balloting”. But in lines 113-115 they wrote “The markets were chosen as sampling sites because farmers from numerous villages bring their products to a central market and people from other villages and towns come to these markets to buy such products”. Based on this information, were the markets randomly selected? Please clarify.

-“Sampling was done in each selected market over two visits at least three months apart.” Was the collection of fecal samples carried out on poultry from the same farmer on each visit?

-The description of how the water was collected for the analyzes is confusing. How was n=100 samples achieved?

-Lines 319 and 321: the terms "genotypically expressed" or "genes were expressed" are not appropriate for the context of analyzing sequenced genomes. Genetic markers - genes that encode resistance to antimicrobials - were detected in the genome of bacteria. What is a possible explanation for the presence of these genes and antimicrobial sensitivity phenotype?

-Figure 3: Is it possible to remove the color from the map, making information about the serovars found in the analyzed markets more visible?

-The title of table 3 is not appropriate: Phenotypic and genotypic resistance to what? I suggest: “Characteristics of Salmonella spp isolated from indigenous poultry in markets located in North Central Nigeria: Phenotypic profile and genotypic markers of antimicrobial resistance, serovars and sequence types.” Moreover, I suggest reporting which samples were collected from water, duck and turkey feces. This information is only in supplementary table S3.

-In the discussion, the authors present Salmonella infection prevalence data (lines 393 - 403). In the present study, did the poutry show signs of infection? The authors must clarify the general condition of the birds selected for sampling: apparently healthy or showing signs of infection.

-Were only the genes (which encode virulence factors) cdtB and SodCI present in the sequenced genomes?

-Review of the English language and typing errors must be carried out.

Minor comments:

-Please add italic in scientific names, and genes

-Can lines 209 - 220 be removed?

-Line 268: please remove one “markets”

Comments on the Quality of English Language

Minor editing of English language required.

Author Response

Responses to Reviewers Comments and Suggestions: Reviewer 1

-In lines 86 and 87, the authors describe that the markets included in the analysis “were randomly selected by balloting”. But in lines 113-115 they wrote “The markets were chosen as sampling sites because farmers from numerous villages bring their products to a central market and people from other villages and towns come to these markets to buy such products”. Based on this information, were the markets randomly selected? Please clarify.

Response: Markets were randomly selected. Lines 113-115 refer to the choice of using markets as sampling sites as against farmer homes. Lines 113-115 should be corrected to read: “ Markets were used as sampling sites because they serve as  congregating points where farmers from nearby villages  bring their products to a central market where people can come and buy such products”

-“Sampling was done in each selected market over two visits at least three months apart.” Was the collection of fecal samples carried out on poultry from the same farmer on each visit?

Response: “No. The study dealt only with the poultry sellers whom the farmers sell to. Faecal samples were taken from the birds that the farmers brought and sold to the poultry sellers who in turn sell to buyers. The farmers that brought their birds to sell may not necessarily have birds to sell next time as selling is mostly done to meet certain family financial obligations”

-The description of how the water was collected for the analyzes is confusing. How was n=100 samples achieved?

Response: The markets have sources where water is collected both for animal and human use. Some markets have 1 or more water sources so samples were collected from those sources. Some are from wells, ground water (boreholes) or streams.

-Lines 319 and 321: the terms "genotypically expressed" or "genes were expressed" are not appropriate for the context of analyzing sequenced genomes. Genetic markers - genes that encode resistance to antimicrobials - were detected in the genome of bacteria. What is a possible explanation for the presence of these genes and antimicrobial sensitivity phenotype?

Response: The ‘silent’ gene concept as explained by Deekshit and Srikumar 2022, means gene detection does not always mean there has to be a corresponding phenotypic expression. Secondly, resistance gene detection may not be sufficient enough to be phenotypically described as resistance (Yee et al 2021).

Figure 3: Is it possible to remove the color from the map, making information about the serovars found in the analyzed markets more visible?

Response: This is noted and colours removed from the map.

-The title of table 3 is not appropriate: Phenotypic and genotypic resistance to what? I suggest: “Characteristics of Salmonella spp isolated from indigenous poultry in markets located in North Central Nigeria: Phenotypic profile and genotypic markers of antimicrobial resistance, serovars and sequence types.” Moreover, I suggest reporting which samples were collected from water, duck and turkey feces. This information is only in supplementary table S3.

Response: Corrections noted and effected

-In the discussion, the authors present Salmonella infection prevalence data (lines 393 - 403). In the present study, did the poutry show signs of infection? The authors must clarify the general condition of the birds selected for sampling: apparently healthy or showing signs of infection.

Response: Apparently healthy birds were sampled. Generally, apparently healthy birds are brought for sale as the ones showing signs of disease might not survive the rigours of transportation and handling.

-Were only the genes (which encode virulence factors) cdtB and SodCI present in the sequenced genomes?

Response: No. Other genes were present but cdtB and SodCI were of interest to us.

-Review of the English language and typing errors must be carried out.

 Minor comments:

-Please add italic in scientific names, and genes

Response: Noted and corrected

-Can lines 209 - 220 be removed?

Response: This has been done

-Line 268: please remove one “markets”

Response: Noted and effected.

Comments on the Quality of English Language

The manuscript was reviewed by a native English speaker

Reviewer 2 Report

Comments and Suggestions for Authors

The manuscript entitled “Antimicrobial Resistance and Phylogenetic Relatedness of Salmonella Serovars in Indigenous Poultry and their Drinking 3 Water Sources in North Central Nigeria”

Thanks for nice work and presentation.

But, some comments should be considered during the revision process;

  1. English editing is important.
  2. The statement “aim of the work” should be mentioned at the beginning of the abstract.
  3. Faecal should be replaced by droppings.
  4. Some abbreviations in the abstract such as “WOAH, PCR, etc.” Should be written as full at the fist mention in the abstract
  5. The abbreviation “MDR” has been mentioned once in the abstract, it should be deleted.
  6. The aim of the study should be the same in the abstract and the introduction.
  7. The different species of poultry from which the samples were collected should be mentioned.
  8. How the water samples were taken from the water sources at the markets include surface water such as streams, ponds or rivers, or ground water such as well and bore holes? Why samples were not collected from the drinkers in front of birds?
  9. Why feed sources were not also collected as samples for Salmonella isolation?
  10. Why authors have measured the antimicrobial resistance in indigenous poultry? What is the benefit?!
  11. Why the serotyping of the obtained isolates were not mentioned despite presence of a table containing the results?
  12. The abbreviations of the media “names” should be deleted. They have been mentioned once.
  13. The genes “invA, cdtB, and SodCI” should be written in a correct way.
  14. A hint about the importance of invA gene in Salmonella spp. could be mentioned in the manuscript.
  15. In conclusion, salmonellosis.

Best regards

Comments on the Quality of English Language

Minor editing is required.

Author Response

Comments and Suggestions for Authors: Reviewer 2

Some comments should be considered during the revision process;

  1. English editing is important.

Manuscript was reviewed and edited by a native English speaker.

  1. The statement “aim of the work” should be mentioned at the beginning of the abstract.

Response: Noted and effected.

  1. Faecal should be replaced by droppings.

Response: Noted and effected.

  1. Some abbreviations in the abstract such as “WOAH, PCR, etc.” Should be written as full at the fist mention in the abstract

Response: Corrections effected.

  1. The abbreviation “MDR” has been mentioned once in the abstract, it should be deleted.

Response: Correction has been effected.

  1. The aim of the study should be the same in the abstract and the introduction.

Response: This is noted and effected.

  1. The different species of poultry from which the samples were collected should be mentioned. Response: Noted and effected in Table 3.
  2. How the water samples were taken from the water sources at the markets include surface water such as streams, ponds or rivers, or ground water such as well and bore holes? Why samples were not collected from the drinkers in front of birds?

Response: Water samples were collected directly into sterile bottles from the wells, boreholes, taps and streams. It was not part of the study design to collect water from drinkers.

  1. Why feed sources were not also collected as samples for Salmonella isolation? Response: It was not part of the study design because feeding and is not organized at the markets. The birds are generally left to scavenge for food at such markets and other times some grains are given to the birds. There may not be any grain left to be used as sample.
  2. Why authors have measured the antimicrobial resistance in indigenous poultry? What is the benefit?! Response: There is generally no available data for indigenous poultry because of the poor veterinary care available, and the close interaction between the farmers and indigenous poultry so it has public health significance, as farmers can be exposed to antimicrobial resistance.
  3. Why the serotyping of the obtained isolates were not mentioned despite presence of a table containing the results? Response: Serotyping was mentioned in Line 178. We have added the methodology.
  4. The abbreviations of the media “names” should be deleted. They have been mentioned once. Response: Correction has been effected.
  5. The genes “invA, cdtB, and SodCI” should be written in a correct way. Response: Correction has been effected.
  6. A hint about the importance of invA gene in Salmonella spp. could be mentioned in the manuscript. Response: Noted and effected.
  7. In conclusion, salmonellosis.
